# Different Dynamic Nodal Properties Contribute to Cognitive Impairment in Patients with White Matter Hyperintensities

**DOI:** 10.3390/brainsci12111527

**Published:** 2022-11-10

**Authors:** Yuanyuan Liu, Shanshan Cao, Baogen Du, Jun Zhang, Chen Chen, Panpan Hu, Yanghua Tian, Kai Wang, Gong-Jun Ji, Qiang Wei

**Affiliations:** 1Department of Neurology, The First Affiliated Hospital of Anhui Medical University, Hefei 230022, China; 2Collaborative Innovation Center of Neuropsychiatric Disorders and Mental Health, Hefei 230032, China; 3Department of Neurology, The Second Affiliated Hospital of Anhui Medical University, Hefei 230601, China; 4The College of Mental Health and Psychological Sciences, Anhui Medical University, Hefei 230022, China; 5Anhui Province Key Laboratory of Cognition and Neuropsychiatric Disorders, Hefei 230032, China

**Keywords:** white matter hyperintensities, cognitive impairment, dynamic topological property, graph theory

## Abstract

White matter hyperintensities (WMHs) are commonly observed in older adults and are associated with cognitive impairment. Although previous studies have found abnormal functional connectivities in patients with WMHs based on static functional magnetic resonance imaging (fMRI), the topological properties in the context of brain dynamics remain relatively unexplored. Herein, we explored disrupted dynamic topological properties of functional network connectivity in patients with WMHs and its relationship with cognitive impairment. We included 36 healthy controls (HC) and 104 patients with mild WMHs (*n* = 39), moderate WMHs (*n* = 37), and severe (*n* = 28) WMHs. The fMRI data of all participants were analyzed using Anatomical Automatic Labeling (AAL) and a sliding-window approach to generate dynamic functional connectivity matrics. Then, graph theory methods were applied to calculate the topological properties. Comprehensive neuropsychological scales were used to assess cognitive functions. Relationships between cognitive functions and abnormal dynamic topological properties were evaluated by Pearson’s correlation. We found that the patients with WMHs had higher temporal variability in regional properties, including betweenness centrality, nodal efficiencies, and nodal clustering coefficient. Furthermore, we found that the degree of centrality was related to executive function and memory, and the local coefficient correlated to executive function. Our results indicate that patients with WMHs have higher temporal variabilities in regional properties and are associated with executive and memory function.

## 1. Introduction

Cerebral small vessel disease (CSVD) arising from disrupted cerebral microvessels is a leading factor in dementia, present in up to 45% of patients [1]. White matter hyperintensities (WMHs), subcortical infarcts, enlarged perivascular space, and microbleeds have been considered common neuroimaging manifestations of CSVD [2], among which the detection rate of WMHs is as high as 72~96% in older adults [3]. Moreover, the increased volume of WMHs and deterioration of white matter microstructure are perceived as predictors of progression to dementia [4]. Increasing evidence has demonstrated that white matter damage mainly involves cognitive impairments in global cognitive function, information processing speed, and executive function [5]. Previous studies have explored the neural mechanisms by which white matter lesions lead to impaired fibril connections and have ignored the fact that non-contiguous brain regions can transmit information. Therefore, previous studies have shown only part of the mechanism of cognitive impairments.

Functional connectivity is viewed as the synchronous activation of blood-oxygen-level-dependent signal observed by the functional magnetic resonance imaging (fMRI) [6]. Numerous pieces of evidence have indicated that the patients with WMHs consisted of within- and between-functional connectivity alterations, which are related to disrupted communication and cognition impairments [7,8]. Graph-theory-based analysis has provided an advanced insight into how WMHs give rise to cognitive impairments through exploring functional brain networks in terms of global properties, such as small-worldness, as well as regional properties, such as the coefficient and nodal degree [9]. Some studies have suggested that the topological properties of functional brain networks in patients with WMHs present increased path length and decreased small-worldness, which influence cognition impairments [10,11].

However, the majority of studies based on static fMRI lacked the exploration of topological properties in the context of brain dynamics. Compared to traditional static functional connectivities (sFC), which is based on the assumption that functional connectivities (FC) patterns are unchanged during scanning, dynamic FC(dFC) can capture temporal changes and thus provide complementary insights into changes in functional brain networks [12]. A graph-theory-based approach applied to dynamic functional connectivity showed that topological dynamical variability in brain networks might offer new insights into the underlying nature of neurodegenerative diseases. For example, the dynamic topological properties of connectivity networks contribute to the assessment of Alzheimer’s disease progression [13] and its prodrome, mild cognitive impairment [14]. Moreover, in schizophrenia, reduced variability in local and global network efficiency is associated with clinical symptoms [15]. Herein, the alteration of dynamic topological properties may provide a novel perspective on cognitive impairments that cannot be detected from sFC.

In this study, we applied widely used automated anatomical labeling (AAL-90) [16] and the sliding window approach to calculate the FC matrix of continuous-time series [17]. Then, the graph-theory method is applied to generate the topological organization of whole-brain connectivity networks and determine differences in topological properties among healthy controls (HCs) and patients with WMHs. Moreover, comprehensive neuropsychological scales were used to assess cognitive function in multiple cognitive domains in patients with WMHs. We hypothesized that patients with WMHs would show increased dynamic regional topological properties, which may be associated with cognitive impairments.

## 2. Materials and Methods

### 2.1. Participants

We recruited 104 right-handed patients from the Department of Neurology of the First Affiliated Hospital of Anhui Medical University. The inclusion criteria were: (1) had macroscopic CSVD-related WMHs upon T2-FLAIR magnetic resonance imaging (MRI); (2) were in the age range of 40 to 80 years old. We did not include patients who were not suitable for MRI scanning or were unable to complete neuropsychological tests (hearing or visual impairments, language or physical movement disorders, etc.). We excluded patients with a history of alcohol addiction, psychiatric disorders, or tumors. Patients who were suspected of having experienced a cardiogenic stroke or with intracranial/extracranial stenosis > 50% or intracranial hemorrhage were also excluded.

Then, patients were divided into three categories according to their Fazekas score: mild WMHs (Fazekas score 1–2; *n* = 39), moderate WMHs (Fazekas score 3–4; *n* = 37), and severe WMHs (Fazekas score 5–6; *n* = 28). The Fazekas score was assessed by a professional neurologist based on the Fazekas scale [18]. Thirty-six HCs were recruited according to the exclusion criteria described above and matched to patients with WMHs in terms of age, sex, and years of education.

The study was approved by the Ethics Committee of the Anhui Medical University. The work described in this article was carried out in accordance with The Code of Ethics of the World Medical Association (Declaration of Helsinki) for experiments involving humans, as well as the Uniform Requirements for Manuscripts submitted to biomedical journals. Prior to data collection, informed consent was obtained from all participants.

### 2.2. Volume of WMHs

The UBO segmentation algorithm was selected to extract the WMH volume (https://cheba.unsw.edu.au/research-groups/neuroimaging/pipeline), which was accessed on 20 September 2021 [19].

### 2.3. Neuropsychological Assessment

A neuropsychological test battery tested for differences in emotional and cognitive performance between groups. The emotional tests included the Patient Health Questionnaire-9 (PHQ-9) [20] and Generalized Anxiety Disorder-7 (GAD-7) [21]. General cognitive function was evaluated by using the Montreal Cognitive Assessment (MoCA) [22], while executive, language, and memory functions were estimated by using the Trail Making Test (TMT), including TMT-A and TMT-B [23], Boston Naming Test (BNT) [24], and Auditory Verbal Learning Test (AVLT) which contain AVLT-study, AVLT-immediate, AVLT-delay, and AVLT-recognition, respectively [25].

### 2.4. MRI Parameters

MRI data were collected by using a GE750w 3.0T MRI system (GE Healthcare, Chicago, IL, USA) at the University of Science and Technology of China. During scanning, participants were instructed to close their eyes, and foam pads were used to minimize head motion. The parameters of acquired sequences were set as follows:

T2 FLAIR: repetition time (TR)/echo time (TE)/inversion time (TI)/flip angle = 8000 ms/165 ms/2000 ms/111°; size of acquisition matrix = 512 × 512; FOV = 256 × 256 mm^2^; 20 continuous slices (thickness = 5 mm with gap = 1 mm); run time = 1 min and 37 s.

Structural, 3D, T1-weighted imaging were set as follows: repetition time (TR)/echo time (TE)/flip angle = 8.16 ms/3.18 ms/12°; number of continuous slices = 188 (thickness = 1 mm with no gap, and voxel size = 1 × 1 × 1 mm^3^); and field of view (FOV) = 256 × 256 mm^2^; run time = 4 min and 43 s.

Rs-fMRI: TR/TE/flip angle = 2400 ms/30 ms/90°; size of acquisition matrix = 64 × 64; FOV = 192 × 192 mm^2^; 46 continuous slices (thickness = 3 mm with no gap, and voxel size = 3 × 3 × 3 mm^3^); run time = 8 min and 41 s.

Enhanced gradient echo T2 star-weighted angiography (ESWAN): TR/TE/flip angle = 52.189 ms/2.856 ms/12°; matrix size = 256 × 256, FOV = 220 × 220 mm^2^; slice thickness = 2 mm with no gap; run time = 5 min and 22 s.

### 2.5. fMRI Data Processing

fMRI images of all participants were preprocessed by DPABI toolbox (http://rfmri.org/dpabi) [26] and SPM12 software (https://www.fil.ion.ucl.ac.uk/spm), which were accessed on 20 September 2021. The first 10 time points were discarded, and the remaining images were used for slice timing and realignment. Head motions were quality-controlled (no more than 3° of rotation or 3 mm of translation in any dimension) during MRI scanning, so no data were discarded. The eligible images were co-registered with functional images and segmented into gray matter (GM), white matter (WM), and cerebrospinal fluid (CSF) [27]. Then, the data were normalized to the Montreal Neurological Institute (MNI) space. Following steps were used to detrend linear regressing covariates (24 head motion parameters, WM and CSF signals) and temporal filtering (0.01–0.08 Hz) [28].

### 2.6. Temporal Network Construction

Based on Dynamic BC toolbox, a sliding-window approach was used to explore the time-varying changes of 90 × 90 pairwise FC matrix by Pearson’s correlations within the nodes clarified by AAL-90 [16,29]. The sliding-windows approach was used to divide the BOLD signal time series, which were extracted from a region of interest (ROI). In this study, the brain was parcellated into 90 ROIs based on the widely used AAL. Then, taking the brain region of interest as the node, we computed the 90 × 90 pairwise functional connectivity matrixes by Pearson’s correlation in individuals’ windows of time series. Based on previous literature recommendations, the time series of MRI scanning was divided into 32 windows (50-TR window = 120 s with 5-TR step size = 12 s) to maintain the stability of functional connectivities and reduce the computational complexity of topological properties [30]. The value of Pearson’s correlations were transformed into z-values by Fisher’s z-transformation.

### 2.7. Dynamic Topological Properties

Given that there is no definite value of network threshold in graph theory analysis, individually obtained temporal FC matrix was converted to binary format with a sparsity threshold of 0.05~0.5 instead of selecting a certain threshold. The definition of sparsity is defined as the ratio of the practical number of existing edges to the maximum possible number of connected edges. Hence, a wide range of sparsity thresholds presented overall value and better estimated the strength of functional connectivities. The negative correlation suggests that these brain regions simply do not form a network and therefore are not considered. Based on GRETNA toolbox (http://www.nitrc.org/projects/gretna/) [31], which was accessed on 21 September 2021, at every sparsity threshold, we applied the graph theory to calculate both global and regional properties in each FC matrix of all sliding windows [32]. In our study, the brain was divided into 90 regions based on AAL, where individual regions were defined as functionally independent nodes and the FC between pairs of regions as edges. Then, we calculated the variance of individual global and regional properties across all sliding windows to obtain a variance value to estimate the variability of topological properties over time, which was further used to test the dynamic topological property differences between groups. The relevant operation steps are illustrated in Appendix A. A small-world network means preferable communication between the nodes in the network. The nodal properties we focused on were degree centrality, betweenness centrality, the nodal clustering coefficient, and nodal efficiency [33,34]. The definition of these topological properties is presented in the Appendix A.

### 2.8. Statistical Analysis

Demographic and neuropsychological data were analyzed by using IBM SPSS Statistics for Windows, version 22.0 (IBM Corp., Armonk, NY, USA). One-way analysis of variance was used to assess differences in age, years of education, and neuropsychological test scores, and the chi-square test was used to determine differences in sex, hypertension, diabetes, hyperlipidemia, smoking history, and drinking history among groups (statistically significant at *p* < 0.05). At the same time, one-way analysis of variance and post hoc analysis were performed for cognition performance among the three WMH groups. The dynamic topological property differences between groups were assessed by one-way analysis of variance (age, sex, and years of education were used as the calibration control variables) and false discovery rate (FDR) correction (statistically significant at a corrected *p* < 0.05). We investigated the relationship between network properties and neuropsychological tests by performing a Pearson’s correlation test and FDR correction (statistically significant at a corrected *p* < 0.05).

## 3. Results

### 3.1. Demographics, Risk Factors, and Neuropsychological Test Results

There were no significant differences in age, sex, or years of education between these groups (Table 1). Compared with HC, patients with WMHs not only showed poor performance in global cognitive function but also in executive, language, and memory functions (Table 2). Moreover, there were significant differences between the three WMH groups (Appendix A), especially in TMT-A and TMT-B.

### 3.2. Dynamic Topological Properties

There were no significant differences in small-world properties between the HC and three WMH groups. However, there were significant differences in the variability of nodal properties (FDR corrected *p* < 0.05) between the four groups. The betweenness centrality differed among the four groups in bilateral rectus gyri, right inferior occipital lobe, right paracentral lobule, right dorsolateral superior frontal gyrus, and left orbital part of the superior frontal gyrus (Figure 1). Nodal efficiencies were altered in the bilateral rolandic opercula (ROL), as well as in the right superior frontal gyrus, right medial orbital part of the superior frontal gyrus, right superior temporal gyrus, and left postcentral gyrus (Figure 2). The nodal clustering coefficient exhibited high variability in bilateral cunei and medial orbital parts of both superior frontal gyrus, as well as in the right superior occipital gyrus, right middle occipital gyrus, right inferior occipital lobe, left superior frontal gyrus, and left supramarginal gyrus (SMG) (Figure 3).

### 3.3. Relationship between Network Properties and Cognitive Function

Correlation analysis revealed that as the temporal variability of the nodal cluster coefficient increased in the left SMG, the TMT-A time gradually increased (r = 0.272, FDR corrected *p* = 0.05) and the AVLT-study score gradually decreased (r = −0.247, FDR corrected *p* = 0.042) in patients with WMHs (Figure 4). Moreover, as the nodal efficiency of the left rolandic operculum increased, the MoCA score gradually decreased (r = −0.281, FDR corrected *p* = 0.012), the TMT-A time gradually increased (r = 0.346, FDR corrected *p* = 0.005), and the TMT-B time gradually increased (r = 0.315, FDR corrected *p* = 0.008) in patients with WMHs (Figure 5).

## 4. Discussion

In this study, the sliding window and graph theory approaches were combined for the identification of alterations in the dynamic topological properties of patients with WMHs. We discovered that patients with WMHs exhibited higher temporal variability than HC in nodal properties, particularly in the superior frontal gyrus and left supramarginal gyrus (SMG), which belong to the cognitive control network (CCN) and default mode network (DMN) [35], but that small-world property did not differ between these groups. Furthermore, altered nodal characteristics were strongly associated with cognitive impairment in patients with WMHs.

Generally, dynamic topological organizations based on fMRI reflect the spatiotemporal volatility of information among brain regions that occurs in continuous sliding windows. Thus, the higher temporal variability of the nodal properties of these regions in patients with WMHs compared to HC in our study suggests asynchronicity in their information transmission with other brain regions. The functional connections in the brain networks influence each other at the spatiotemporal level, and intricate brain functions depend on integration and separation among functionally specialized neural circuits [36]. Asynchronicity of the brain regions may impede the transmission of information within and/or between brain networks [37]. Conceptually, temporal nodal properties are commonly measured in terms of localized temporal information transmission. Nodal properties refer to the degree of information transmission to other nodes or fault tolerance of a priority node in the subnetwork [32]. Similarly, our results indicated that patients with WMHs exhibited higher temporal variability than HC in terms of nodal properties, particularly in the superior frontal gyrus and left supramarginal gyrus (SMG), which belong to the CCN and DMN [35]. Thus, it seems that higher variability in the dynamic nodal properties of the brain networks reduces their anatomical constraints, and spatially distant functional connections to these pivotal brain regions may lead to further cognitive impairments. However, the small-world property did not differ between these groups. The small-worldness metric σ illustrates this relationship using data from multiple networks [38] may not be as sensitive to changes in global small-worldness. The network presented in this work is relatively small (*n* = 90) in a regime where σ may be close to its critical value of 1. Because of this, the dynamic variability in small-worldness may not be sufficient to detect differences associated with WMH burden. Moreover, the small world is a stable topology, so it is not prone to present temporal variability in a relatively short period. Moreover, some dynamic topological properties are only produced in the group with severe WMHs. It may be that although the topological structure of patients with mild-to-moderate WMHs changes during the scan, the dispersion of the changes is not large and is still within the compensation range. Thus, it is not that different from the dynamic change in HC. This is consistent with the lack of between-group differences in the small-world property.

Furthermore, we discovered that variation in dynamic nodal topological organizations mainly occurred in the CCN and DMN and was closely associated with executive and memory function in patients with WMHs, which was mainly observed in SMG and ROL. The SMG resides in the inferior parietal lobule and is a node of the DMN, establishing complex functional connections between different networks for the regulation of executive functions [35]. Moreover, the SMG can regulate bottom-up attentional orienting and memory retrieval [39]. The rolandic operculum, regarded as the main node of the CCN, mediates the encoding and extraction of memories [40] and is involved in information processing speed [41], which is a key subdomain of cognitive function and is now commonly treated as a pivotal diagnostic indicator for neurocognitive disorders [42].

Therefore, higher fluctuations in the dynamic brain networks may result in a disconnect among information processing systems [43]. This is also consistent with patients with WMHs showing impaired executive function and memory abilities [44]. From the perspective of dFC, our study further complements the findings of previous sFC studies that cognitive impairment is associated with impaired node properties [45].

Some limitations that may affect the results should be taken into account. First, in this study, our study sample was small; therefore, the results should be considered preliminary, and causality remains to be determined with larger samples and long-term follow-up studies. Second, a further multicenter study should be considered to minimize the influence of regional cultural differences on cognitive performance in a simple cross-sectional study.

## 5. Conclusions

In conclusion, our results indicate that the patients with WMHs had higher temporal variability in regional properties, including betweenness centrality, nodal efficiencies, and nodal clustering coefficient. Furthermore, most of the variable nodal properties are located within the DMN and CCN and are associated with executive and memory functions.

## Figures and Tables

**Figure 1 brainsci-12-01527-f001:**
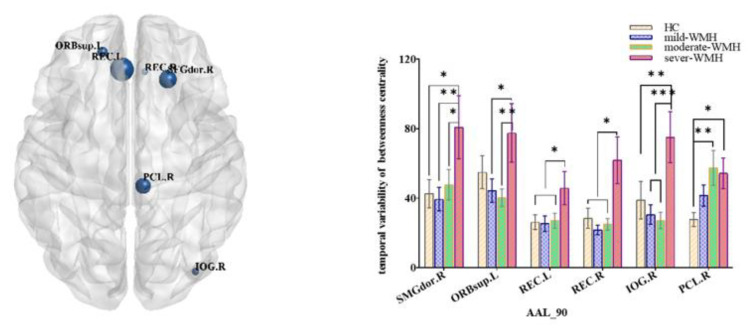
Group comparisons of the temporal variability of betweenness centrality. Right rectus gyrus (REC.R), right inferior occipital lobe (IOG.R), right paracentral lobule (PCL.R), right dorsolateral superior frontal gyrus (SFGdor.R), left rectus gyrus (REC.L), and left orbital part of the superior frontal gyrus (ORBsup.R). *** significant at 0.001 level, ** significant at 0.01 level, and * significant at 0.05 level (2-tailed).

**Figure 2 brainsci-12-01527-f002:**
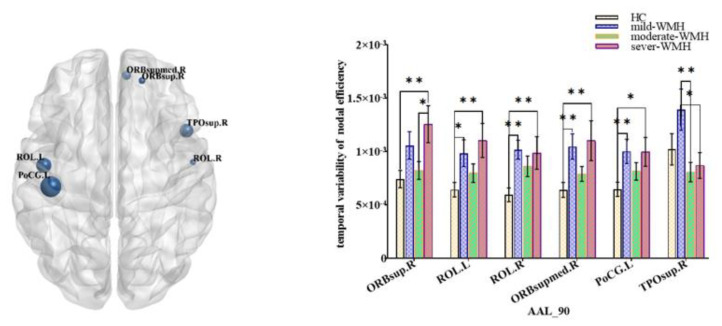
Group comparisons of the temporal variability of nodal efficiency. Right rolandic operculum (ROL.R), right orbital part of superior frontal gyrus (ORBsup.R), right medial orbital part of superior frontal gyrus (ORBsupmed.R), right temporal pole: superior temporal gyrus (TPOsup.R), left rolandic operculum (ROL.L), and left postcentral gyrus (PoCG.L). ** significant at 0.01 level, and * significant at 0.05 level (2-tailed).

**Figure 3 brainsci-12-01527-f003:**
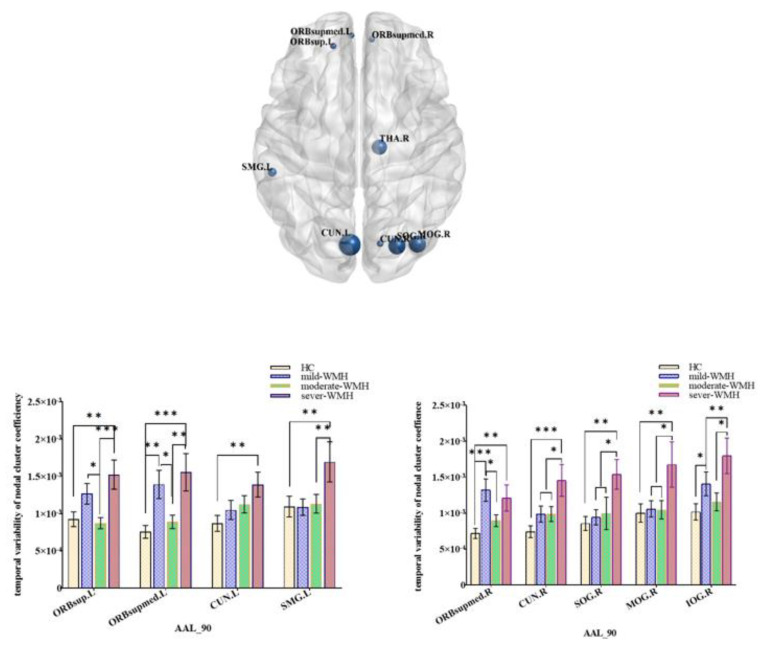
Group comparisons of the temporal variability of the nodal cluster coefficiency. Right cuneus (CUN.R), right medial orbital part of superior frontal gyrus (ORBsupmed.R), right superior occipital gyrus (SOG.R), right middle occipital gyrus (MOG.R), right inferior occipital lobe (IOG.R), left cuneus (CUN.L), left medial orbital part of superior frontal gyrus (ORBsupmed.L), left orbital part of superior frontal gyrus (ORBsup.L), and left supramarginal gyrus (SMG.L). *** significant at 0.001 level, ** significant at 0.01 level, and * significant at 0.05 level (2-tailed).

**Figure 4 brainsci-12-01527-f004:**
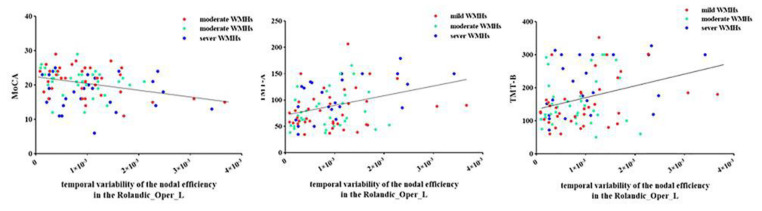
Correlations between the temporal variability of nodal efficiency in the left rolandic operculum and MoCA score, TMT-A time, and TMT-B time in patients with WMHs. WMHs, white matter hyperintensities; MoCA, Montreal Cognitive Assessment; TMT, Trail Making Test.

**Figure 5 brainsci-12-01527-f005:**
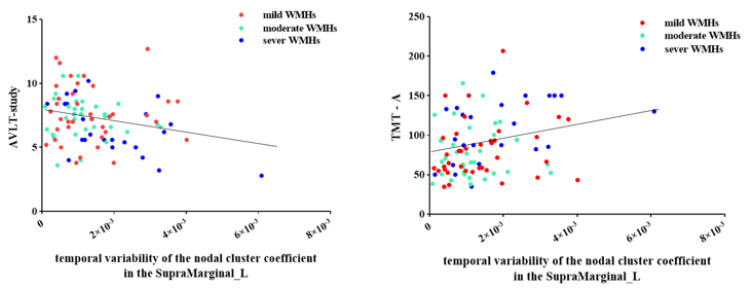
Correlations between the temporal variability of nodal cluster coefficiency in left supramarginal gyrus and TMT-A time and the score of AVLT-study in patients with WMHs. WMHs, white matter hyperintensities; AVLT, Chinese Auditory Learning Test; TMT, Trail Making Test.

**Table 1 brainsci-12-01527-t001:** Demographic and neuroimaging manifestations.

	HC (*n* = 36)	Mild WMHs (*n* = 39)	Moderate WMHs (*n* = 37)	Severe WMHs (*n* = 28)	F/χ^2^	*p* Value
Age, years (mean (SD))	60.58 ± 5.98	63.77 ± 8.23	65.08 ± 10.03	65.04 ± 6.90	2.424 ^b^	0.068
Female, *n* (%)	18 (50.0%)	16 (41.0%)	14 (37.8%)	16 (57.1%)	3.011 ^a^	0.390
Education, years (mean (SD))	9.31 ± 3.32	8.41 ± 4.16	8.00 ± 3.84	7.75 ± 3.88	1.083 ^b^	0.359
Hypertension, *n* (%)	8 (22.2%)	21 (53.8%)	22 (59.5%)	21 (75.0%)	19.568 ^a^	<0.001 ***
Diabetes, *n* (%)	2 (5.6%)	8 (20.5%)	7 (18.9%)	4 (14.3%)	3.905 ^a^	0.272
Hyperlipidemia, *n* (%)	8 (22.2%)	10 (25.6%)	9 (24.3%)	8 (28.6%)	0.356 ^a^	0.949
Smoking history, *n* (%)	9 (25.0%)	18 (46.2%)	15 (40.5%)	11 (39.3%)	3.808 ^a^	0.283
Drinking history, *n* (%)	12 (33.3%)	9 (23.1%)	16 (43.2%)	12 (42.9%)	4.346 ^a^	0.226
Fazekas	0.00 ± 0.00	1.64 ± 0.49	3.59 ± 0.50	5.43 ± 0.50	958.855 ^b^	<0.001 ***
WMHs volume	/	11,457.35 ± 12,493.05	18,067.56 ± 10,873.84	30,658.380 ± 12,033.91	21.696 ^b^	<0.001 ***
Lacunes	0.00 ± 0.00	0.51 ± 0.91	0.89 ± 1.29	0.79 ± 1.23	5.626 ^b^	0.001 **
Microbleeds	0.31 ± 0.59	2.26 ± 3.61	1.50 ± 2.35	6.29 ± 14.96	3.389 ^b^	0.021 *

^a^ Chi-square test; ^b^ one-way analysis of variance. Abbreviations: HC, healthy control; WMHs, white matter hyperintensities; SD, standard deviation. *** significant at 0.001 level, ** significant at 0.01 level, and * significant at 0.05 level (2-tailed).

**Table 2 brainsci-12-01527-t002:** Neuropsychological tests.

	HC (*n* = 36)	Mild WMHs (*n* = 39)	Moderate WMHs (*n* = 37)	Severe WMHs (*n* = 28)	F	*p* Value
MoCA	22.33 ± 3.09	21.16 ±4.32	21.06 ± 3.79	17.81 ± 4.91	6.850	<0.001 ***
AVLT-study	8.15 ± 1.59	7.63 ± 2.24	7.31 ± 1.52	6.55 ± 2.00	3.776	0.012 *
AVLT-immediate	9.40 ± 2.30	8.17 ± 3.84	7.88 ± 2.65	6.17 ± 3.27	5.320	0.002 **
AVLT-delay	9.06 ± 2.41	7.95 ±3.76	7.31 ± 2.67	6.50 ± 3.13	3.731	0.013 *
AVLT-recognition	13.63 ± 1.33	13.53 ± 3.68	13.16 ± 1.87	11.13 ± 3.62	4.734	0.004 **
TMT-A	63.34 ± 24.76	80.72 ± 33.12	79.12 ± 33.73	105.82 ± 37.72	8.233	<0.001 ***
TMT-B	129.65 ± 44.44	155.35 ± 70.69	158.16 ± 73.16	207.55 ± 81.24	6.301	0.001 **
TMT(B-A)	66.31 ± 34.68	68.68 ± 51.19	78.57 ± 50.56	102.24 ± 57.58	3.074	0.030 *
BNT	13.94 ± 0.98	12.89 ± 1.54	13.25 ± 1.68	12.74 ± 1.40	4.797	0.003 **
PHQ-9	3.31 ± 4.78	3.74 ± 4.56	5.00 ± 4.62	7.39 ± 5.81	4.272	0.006 **
GAD-7	2.40 ± 3.24	2.61 ± 4.10	2.81 ± 3.43	5.29 ± 5.48	3.282	0.023 *

Data are mean ± SD. Abbreviations: HC, healthy control; WMHs, white matter hyperintensities; SD, standard deviation; MoCA, Montreal Cognitive Assessment; AVLT, Chinese Auditory Learning Test; TMT, Trail Making Test; BNT, Boston Naming Test; PHQ, Patient Health Questionnaire; GAD, Generalized Anxiety Disorder. Volumes are in cubic millimeters. *** significant at 0.001 level, ** significant at 0.01 level, and * significant at 0.05 level (2-tailed).

## Data Availability

Not applicable.

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
