# Peer review of "Different Dynamic Nodal Properties Contribute to Cognitive Impairment in Patients with White Matter Hyperintensities"

_brainsci, 2022, doi:10.3390/brainsci12111527_

Round 1
Reviewer 1 Report
The subject is interesting for the scientific and clinical community and deserves a reformulation to be ready for publication.
Method section:
-the demographic recruitment is confusing because of the mix of some information on the final sample. Please, focus in this section only on the criteria used to include or exclude volunteers from each group.
-The authors did not include the volume number information of the functional MRI data.
Results section:
-the results were well represented by the figures, but the results were not better described in detail, and important relationships between the results were not discussed. The authors could have better explored the results and their correlations with more specific comparisons with the extremes and their relevance.
-I suggest exploring the significant results of cognitive data, showing the post hoc analyses of Table 2. In Figures 4 and 5, I suggest showing the WMHs groups in different colors.
Reviewer 2 Report
General Summary
Liu and colleagues report on a sample of individuals with white matter hyperintensities (WMH) of varying severities and healthy controls. The authors expand on previous work comparing resting state functional connectivity in individuals with WMH to healthy controls by examining dynamic functional connectivity and connecting it to different domains of cognition. The authors find that not only do participants with WMH show cognitive impairments relative to healthy controls, they also show differences in the nodal properties of select regions that presumably reflect differences in temporal stability of these regions. Several brain-behavior correlations within participants with white matter disease were also identified.
General Comments
The manuscript clearly offers a step forward in terms of understanding how WMH impacts functional connectivity. However, a critical piece seems to be missing. The crux of the paper is that dynamic functional connectivity may offer further insight than static functional connectivity, but the methods section does not indicate how temporal variability of graph theory measures was taken into account. Moreover, the figure labels don’t make it clear that readers are looking at variability of these measures over time. This severely detracts from the manuscript and makes it difficult to follow the conclusions of the article. It is absolutely key for the authors to make it clear how this was done.
Overall, the methods section can be really beefed up. Not only should that alleviate the above issue, but it would also really help the readers through what was done. The specific comments offer suggestions as to where this can be done.
There is a good amount of causal language strewn throughout the paper. Consider removing.
Specific Comments
Abstract
Please just specify that the 104 patients were grouped into different severities of WMH. It’s a bit misleading as written.
Intro
P1L44-P2L47: It would be helpful to break up this sentence. This also seems like a hard statement to make given the data presented. I’m not sure the authors intend to determine whether white matter abnormalities or functional connectivity contribute more to cognitive decline (unless I am mistaken). Consider simply stating that altered functional connectivity, due to or in parallel with white matter abnormality, may provide insight into cognitive impairment.
P2L49-L52: “…WMHs-related cognitive impairment is not only caused by the disruption of functional connectivity…” Again, I’m not sure any causation is determined in the reference here. Perhaps replace “caused” with “associated.”
P2L49-52 (same section as above): Please spell out the names of these networks the first time. More importantly, why are only these three networks mentioned? White matter hyperintensities afflict the whole brain, no? Perhaps this is what previous studies have shown (but see Kantarovich et al., 2022)? And if so, this could use some unpacking. Please also amend the words “which are responsible for…” with something like “are involved in.”
Kantarovich, K., Mwilambwe-Tshilobo, L., Fernández-Cabello, S., Setton, R., Baracchini, G., Lockrow, A. W., Spreng, R. N., & Turner, G. R. (2022). White matter lesion load is associated with lower within- and greater between- network connectivity across older age. Neurobiology of aging, 112, 170–180. https://doi.org/10.1016/j.neurobiolaging.2022.01.005
P2L54: “…how WMHs give rise to clinical symptoms…” What clinical symptoms? So far you have only mentioned cognitive impairment.
P2L59-61: This seems like a strange place to put this sentence as you haven’t yet suggested why that would be interesting. Perhaps move to the first sentence of the next paragraph?
P2L73: What do you mean by applied AAL? Please specify that AAL was used to identify ROIs from the networks of interest.
P2L79-80: Please specify what you hypothesize “abnormal dynamic topological properties” to be and which measures you expect to see changes in. Rather than “abnormal,” consider saying “compared to healthy controls.” What would support of your hypothesis suggest?
Methods
P3P99-100: Please mention who provided Fazekas scores (i.e., authors, radiologist, automatic?).
Please expand on the Neuropsychological Assessment section. What are the “main cognitive domains?” What is the Patient Health Questionnaire? What is the motivation for including an anxiety scale? What dependent variables will be used for each measure? Please also refer to Table 2 here so readers know to look for scores there. For TMT, please report B-A (latency) as well.
MRI parameters: please specify what functional data were collected (i.e., resting state, how many runs, how long were the runs)
Dynamic topological properties: How were networks defined? Does AAL organize the 90 nodes into networks? Apologies if I missed it, but this seems important to understand P3L144 (“…we calculated the topological properties of the brain network…”
P4L147-148P: Why was this sparsity threshold chosen? What about negative connections?
Statistical testing: A follow-up to the above comment on network definition: I don’t quite follow how graph theory measures were tested. Were ANOVAs and correlations done on each of the 90 regions? If that is the case, please adjust the language in the methods accordingly to avoid confusion. Then it would be fitting to discuss what networks these regions are typically involved in. For the correlations, that is a lot of tests. Consider using a more stringent correction (i.e., Bonferroni). Were correlations conducted in the full group or only WMH participants (results section makes it seem like just WMH group)? Please specify.
I’m also missing the time component here. It could be because I’m not entirely familiar with the software, but it could use clarifying either way. How is variability of the graph theory measures being assessed? Is window included as a within-subject variable in the ANOVAs? Or is some additional measure derived to reflect variability of each measure? Or are the measures themselves measures of variability?
Results
Neuropsychological testing: It would be helpful to know not just whether there was a difference between the four groups, but also whether there were differences between the 3 WMH groups. Please include follow-up pairwise t-test results.
Dynamic topological properties:
P5L192-196: Please specify whether this result was overall or specific to one group.
Again, I’m missing the time component here. The figure labels don’t indicate whether we’re looking at mean values or variability of values, so it’s hard to really understand the stability over time.
Correlations: Can you color code Figures 4& 5 by WMH group? This would aid in interpretation. Please also add more descriptive x axis labels to convey what graph theory measure is being correlated with cognition.
Discussion
What might it mean that there were differences in nodal properties but not small-world properties?
Assuming that the values presented really do reflect variability (see above comments) I’d like to see some more interpretation of the nuance here. In some results, it seems that severe WMH really drives the group difference (Fig 1), whereas in others, simply being categorized as having WMH is enough to show a difference from controls. Why might this be the case for these measures and regions?
P9L284-285: This seems like a hard statement to make given that you don’t look at negative correlations.
Overall
Please be sure to spell out all abbreviations before using them: DMN, FPCN, DAN, sFC, CCN, etc…
Round 2
Reviewer 1 Report
I am satisfied with the changes made by the authors and I consider that the manuscript is suitable for publication.
Author Response
Thank you
Reviewer 2 Report
Please see attached.
